# Approaches to CNS Drug Delivery with a Focus on Transporter-Mediated Transcytosis

**DOI:** 10.3390/ijms20123108

**Published:** 2019-06-25

**Authors:** Rana Abdul Razzak, Gordon J. Florence, Frank J. Gunn-Moore

**Affiliations:** 1Medical and Biological Sciences Building, School of Biology, University of St Andrews, St Andrews KY16 9TF, UK; rar5@st-andrews.ac.uk; 2Biomedical Science Research Centre, Schools of Chemistry and Biology, University of St Andrews, St Andrews KY16 9TF, UK; gjf1@st-andrews.ac.uk

**Keywords:** CNS-targeted drug delivery, Blood-brain barrier, Receptor-mediated transcytosis, Transient BBB disruption, Efflux-pump inhibition, Ring-opening metathesis polymerization

## Abstract

Drug delivery to the central nervous system (CNS) conferred by brain barriers is a major obstacle in the development of effective neurotherapeutics. In this review, a classification of current approaches of clinical or investigational importance for the delivery of therapeutics to the CNS is presented. This classification includes the use of formulations administered systemically that can elicit transcytosis-mediated transport by interacting with transporters expressed by transvascular endothelial cells. Neurotherapeutics can also be delivered to the CNS by means of surgical intervention using specialized catheters or implantable reservoirs. Strategies for delivering drugs to the CNS have evolved tremendously during the last two decades, yet, some factors can affect the quality of data generated in preclinical investigation, which can hamper the extension of the applications of these strategies into clinically useful tools. Here, we disclose some of these factors and propose some solutions that may prove valuable at bridging the gap between preclinical findings and clinical trials.

## 1. Introduction

Decades of dedicated efforts have led to scientific advances in understanding the physiology of many diseases and development of a wide array of therapeutic materials. To fulfil their intended purpose, therapeutic materials must have adequate pharmacokinetics and pharmacodynamics that can allow them to accumulate in the diseased site in their therapeutically effective concentration. Advances in the treatment of infections or malignancies residing in peripheral tissues are growing at a faster pace compared to their CNS-associated equivalents. In large part, this phenomenon is due to the inability of the therapeutic material to cross brain barriers from the systemic circulation into brain parenchymal tissues. In this review, we describe the anatomical structure of brain barriers with a focus on the major contributor to the brain’s strict permeability. We review excerpts of CNS drug delivery approaches available for clinical use or still undergoing clinical or preclinical trials. We also highlight the principles underlying their mechanism of action and bring out some of their advantages and limitations. Finally, we attempt to follow their progress as they weave their way from preclinical investigation to clinical use.

Targeted drug delivery has found applications in the diagnosis and treatment of many diseases, such as cancer [1], diabetes [2] and neurodegenerative diseases [3]. As yet, neurodegenerative diseases remain an area where targeted drug delivery is most needed, since surgical treatments are not always an option and a treatment at a molecular level is needed. Additionally, intravenously injected neurotherapeutic materials not only have to evade en route biological barriers, such as solubility in the blood stream, stability against enzymatic degradation and phagocytosis, but also barriers interposed between the systemic circulation and brain tissues [4]. Barriers of the central nervous system (CNS) are made of specialized cells that lie at the interface between blood and nervous tissues, forming the blood-brain barrier (BBB), and between blood and cerebrospinal fluid, forming the blood-cerebrospinal fluid barrier (BCSFB) Figure 1 [5,6,7,8,9]. These specialized cells express transmembrane proteins with which they can seal off the intercellular space protecting brain tissues from micro-organisms, toxic compounds and fluctuations in the blood stream that can disrupt synaptic transmission allowing the brain to efficiently perform its vital functions [5,10]. The BBB is considered as the primary contributor to the brain’s strict permeability due to its larger surface area and its faster blood flow rate in comparison to BCSFB [8]. BBB cells are the endothelial cells that line brain vasculature, while BCSFB cells are the choroid plexus epithelium cells that line the cerebral ventricles and the arachnoid epithelium that line the brain vasculature in the subarachnoid space [5].

## 2. Structural Components of the BBB

The major contributor to the BBB occlusion and strict permeability are the tight junctions (TJs) that seal off the intercellular space between the endothelial cells (ECs) lining the cerebral microvessels [11,12,13]. TJs (e.g., claudins, occludins and junctional adhesion molecules) are transmembrane proteins that bind extracellularly to identical transmembrane proteins in adjacent ECs and intracellularly to the actin component of the filamentous cytoskeleton [11,12,13,14,15]. The occlusion and strict permeability of the BBB is maintained and regulated by mural and glial cells in response to neuronal activity and needs [11,12,13,14,15,16,17,18]. Mural cells, pericytes and smooth muscle cells, embedded in an extracellular perivascular matrix termed basement membrane, control the diameter of capillaries, thus regulating blood flow and immune cell infiltration [10,13,19]. Astrocytes, a component of glial cells, ensheath the ECs, mural cells and the extracellular matrix, supporting blood vessels’ architecture and maintaining BBB integrity [5,10,11,20]. Astrocytes secrete chemical reagents that modulate BBB permeability in response to neuronal signals, providing the cellular linkage between neuronal circuitry and blood vessels [5,10,11,20]. Together, the TJs, pericytes and astrocytes form the physical components of the BBB.

Due to the absence of vascular fenestrae, molecules (e.g., nutrients, electrolytes, metabolic waste and potentially therapeutic material) are translocated across the BBB’s vascular endothelium via specialized transporters (Figure 2) [5,10,16,21,22,23]. Nutrients, such as glucose, amino acids, peptides and proteins, are actively transported down their concentration gradient from the vascular lumen to the brain parenchyma via carriers or receptors expressed at the luminal and abluminal plasma membranes of the BBB’s ECs [5,10,22]. Influx of electrolytes, e.g., Na^+^ and K^+^, is regulated by ion pumps residing on the abluminal surface of EC membranes [16,24]. Potentially toxic substances, such as amyloid-beta, a peptide implicated in the pathology of Alzheimer’s disease, are cleared from the brain parenchyma by receptors expressed at the abluminal membrane [10,16]. Small molecules, e.g., CO_2_, O_2_ and lipophilic molecules, with molecular weights <400 Da, are able to passively diffuse through the lipid bilayer membranes of ECs [5,10,16,22,23].

The passive diffusion of small lipophilic molecules through the lipid bilayer across the BBB, in most scenarios, is counteracted by enzymatic metabolism or active efflux from vascular endothelium into the blood [5,10,25]. Active efflux is accomplished by ATP-binding cassette transporters (ABC), such as P-glycoprotein (P-gp), breast cancer resistance proteins (BCRPs) and multidrug resistance-associated proteins (MRPs) [10,16,22,23]. ABC transporters are transmembrane proteins that use the hydrolysis of adenosine triphosphate (ATP) as the driving force for the translocation of their substrate against the substrates’ concentration gradient [26]. Albeit essential to protecting brain homeostasis and vitality against foreign molecules, efflux transporters pose additional hurdles in the delivery of neurotherapeutics across the BBB due to their broad substrate scope [27,28].

### 2.1. Strategies for Delivery of Therapeutics Across the BBB

Technical strategies for the delivery of therapeutics across the BBB, as we shall see, are broadly divided into two main categories—delivery beyond and through the BBB. Delivery beyond the BBB bypasses the BBB altogether by injecting the therapeutic material directly into the brain tissues. Delivery through the BBB can be achieved either by enhancing the physicochemical properties of the therapeutic material, inhibition of efflux pumps, transient disruption of the BBB or by exploiting endogenous transport mechanisms employed by ECs for the translocation of nutrients. In this section we will present examples of both approaches and highlight the underlying principles and some of the advantages and disadvantages.

#### 2.1.1. Drug Delivery Beyond the BBB

##### Intraparenchymal Drug Delivery

Intraparenchymal drug delivery is the direct injection of therapeutic material in the brain interstitium (Figure 3) [29,30,31]. Brain interstitium is the region lying within the extracellular and extravascular space in brain parenchyma [29,31]. The space is occupied with interstitial fluid, which consists of water, salts, sugars, fatty acids, amino acids, hormones, neurotransmitters and other nutrients [29,30,31]. Injecting the therapeutic material directly in the brain interstitium allows it to reach different regions in the brain by traversing the interstitial space [29,30,31]. Due to its interspersion with cells and capillaries, the interstitium is an inhomogeneous and anisotropic environment and can often be altered by a disease’s pathological state [29,30,31]. In practical terms, this means that injection is associated with inevitable, and sometimes permanent, damage to nearby tissues [29]. Additionally, the procedure is fraught with technical complications, such as inadequate infusate diffusion, and in some cases, backflow [29].

##### Intraventricular and Intrathecal Drug Delivery

The therapeutic material is directly administered in one of the ventricles (intraventricular) or the spinal canal (intrathecal) as time-separated boluses via an internally implanted reservoir or a programmed pump (Figure 4) [32,33,34,35]. The therapeutic material reaches the brain and spinal cord tissues through the CSF [32,33,34,35]. Replenishment of the therapeutic material can be achieved by topping up the reservoir or replacing the pump [32,33,34,35]. The installation of the catheter and the reservoir or the pump is performed surgically, thus, as with all surgical manipulations, there is a risk of tissue damage, accompanied with a loss of function, and tissue infection with the added recovery time of the patient [32,33,34,35]. The extent of such risks can be reduced by careful planning of the operation, use of orthogonal technologies that can help visualize the route of implantation, and antibiotic prescription, if necessary [32]. In some cases, intraventricular or intrathecal drug administration is the most cost-effective approach for the treatment of chronic pain secondary to brain or spinal cord injury by sustaining a prolonged and stable state of analgesia without developing addiction or severe side effects, thereby improving patients’ quality of life [32].

##### Intranasal Drug Delivery

Therapeutic material administered intranasally can reach the CNS via three transport routes; the olfactory pathway, the trigeminal pathway and the systemic circulation pathway [36,37]. The olfactory and trigeminal pathways bypass the BBB by delivering the therapeutic material from the nasal cavity to the brain tissue via passive or active transport mechanisms along the olfactory nerves or trigeminal nerve branches that innervate the nasal cavity (Figure 5) [36,37]. As for the systemic circulation pathway, the therapeutic material must diffuse through the mucus, cross the nasal epithelium by passive or active transport, enter the venous blood flow to reach the CNS, then overcome the BBB [36,37]. The olfactory and trigeminal pathways are the direct nose-to-brain transport routes in this scenario and are much less invasive than the intraparenchymal and intraventricular routes. However, these pathways are limited by the small amount of therapeutic material that can successfully reach the brain tissue, relative to the amount administered [36,37].

#### 2.1.2. Drug Delivery Through the BBB

##### Optimizing the Physicochemical Properties of Therapeutic Materials

To fulfil its intended purpose, a drug must exhibit high potency and selectivity towards the biological target, as well as reach a concentration above an experimentally determined threshold within brain tissue [25]. Drug concentration in the brain, C_br_, is influenced by the concentration of the drug in its free form in the blood stream, as described by Equation (1), where K_in_ is the unidirectional influx constant [38,39].

(1)$$C_{br}=K_{in}\times AUC_{0}{}^{T}$$

The concentration of the therapeutic material in its free form in the blood pool, expressed as the area under the plasma concentration-time curve (AUC_0_^T^), is affected by its pharmacokinetic properties [25]. Pharmacokinetic refers to how the host’s body acts on the therapeutic material once it’s administered by absorption, distribution, metabolism and excretion processes (Figure 6) [25,38,39]. Therefore, optimizing the physiochemical properties of drug candidates to evade these processes has become an integral part of the drug discovery paradigm [25,40,41]. In this context, the most detrimental physiochemical properties are lipophilicity, hydrogen bonding, molecular flexibility and molecular volume [25,40,41]. Drawn from literature precedents in this area, lipophilicity, hydrogen bonding and molecular flexibility follow a parabolic curve that exists between these properties and the drug’s distribution in the brain [25]. Increasing the lipophilicity improves the solubility of drug candidates in ECs’ lipid membranes but it can negatively affect the drug’s absorption, and consequently its AUC_0_^T^, due to insufficient solubility in the blood stream [25,42,43]. Increasing the number of heteroatoms capable at forming hydrogen bonds may resolve drug absorption limitations as a result of increased hydrophilicity [25]. Nonetheless, it can introduce other issues, such an increase in free energy of desolvation due to the enthalpy penalty associated with the polar groups desolvation, which is a process the drug must undergo as it approaches the ECs’ lipid membranes and attempts to initiate a dipole interaction with its components [25,44]. Additionally, increased polarity and/or lipophilicity may enhance drug binding to serum proteins via H-bonding and lipophilic interactions, respectively, which, in turn, reduces its AUC_0_^T^, and therefore, C_br_ [25,45]. Molecular flexibility, induced by rotatable bonds, allows the drug molecule to wiggle through the long chains of the membrane lipids [25,46]. However, excessive molecular flexibility may have the opposite effect if the drug molecule can adopt an array of conformations among which are bulky geometries [25,46]. As for molecular volume which is a function of molecular weight and structure including all accessible conformations the drug molecule can adopt under the physiological conditions, the more compact it is the greater is the likelihood of the drug traversing the BBB by passive diffusion through its ECs’ LBL membranes [25]. Retrospective analysis of CNS and non-CNS active drugs that had entered at least phase II of clinical trials determined the general physicochemical properties required for BBB penetration and has guided the synthesis of novel analogues [25]. The growth of computing power allowed the development of computational algorithms that can predict the physicochemical and pharmacokinetic properties of drug candidates and their capacity to bypass the BBB [25].

##### Inhibition of Efflux Proteins

Inhibition of efflux pumps allows systemically administered drugs that can passively cross the BBB to accumulate in the brain and reach their therapeutically effective concentration [47,48,49,50,51]. The inhibitor, typically co-administered with the drug, can attenuate the activity of efflux pumps by competitively or allosterically blocking the drug binding site or by interfering with ATP hydrolysis that drives the efflux of substrates against their concentration gradient (Figure 7) [48,51]. In pre-clinical studies, co-administration of elacridar, a P-gp allosteric modulator, with paclitaxel, a chemotherapeutic agent, was shown to increase the latter’s brain uptake in mouse models relative to matched controls [50,51]. Efflux transporters are ubiquitously expressed in several human tissues, such as kidneys and liver, therefore the pharmacokinetic fate of therapeutic material co-administered with efflux inhibitor is often unpredictable [52,53]. Whereas down-regulation of efflux transporters’ activity can enhance brain uptake of the therapeutic material, it can also increase brain uptake of potentially toxic materials. Therefore, the risks associated with long-term therapy utilizing this approach must be carefully weighed.

##### Transient Disruption of the Blood–Brain Barrier

Hyperosmotic Disruption of the BBB

Intracarotid injection of a hyperosmotic solution such as arabinose or mannitol prior to intraarterial administration of chemotherapeutic agents was shown to substantially raise the latter’s concentration in the brain parenchyma [54,55]. The hyperosmotic solution causes water to be withdrawn out of the ECs into the vascular lumen, resulting in shrinkage of the ECs (Figure 8) [54,55]. Shrinkage of the ECs causes their cytoskeleton to horizontally contract, whose effect propagates through the cytoskeleton’s actin component, TJs’ adapter proteins, and finally reaches the TJs proteins [54,55]. The downstream effect is an increased bulk fluid flow across the tight junctions into brain parenchyma [54,55]. The transient disruption induced by this hyperosmotic method lasts for a short period of time—15 min to 4 h—the length of which can be modulated by varying hyperosmotic solution concentration and infusion time [54,55]. The hyperosmotic disruption technique is non-specific with respect to the cerebral and brain tissue it can affect. The hyperosmotic solution can circulate in the blood stream and induce its affect in several brain and, potentially, body regions [55]. Despite its low morbidity rates, hyperosmotic disruption of the BBB is associated with transient cerebral edema due to increased bulk fluid influx. This can severely aggravate neurological defects, especially if such treatment regimen is to be prescribed repeatedly [54].

Microbubble-Assisted Focused Ultrasound Irradiation (MB-FUS)

Image-guided focused ultrasound (FUS) emerged as an alternative approach for the induction of transient BBB disruption after it had for been tested for a long time as a non-invasive thermal ablation technique for the treatment of brain tumors [56,57,58]. Early investigations of FUS-induced BBB disruption and its bioeffects on brain tissue reported the occurrence of tissue damage that manifested itself as lesions or necrosis due to exposure to high frequencies of FUS [59,60,61,62,63]. Intravenous injection of preformed microbubbles prior to FUS exposure prevented tissue damage by attenuating the effective FUS power levels [64,65,66,67]. The microbubbles, which are typically commercially accessible lipid or albumin shells encapsulating gaseous material, such as Optison, harness the acoustic power and concentrate it to the blood vessel walls maximizing its efficiency, and thus, lowering the threshold of effective power levels (Figure 9) [64,65,66,67]. Circulating micro-bubbles move in the direction of the FUS wave propagation and are brought in close contact with the vessel’s ECs [64,65,66,67]. Depending on their size and the acoustic power level, microbubbles can oscillate, micro-streaming the medium surrounding them, expand, or collapse, exerting mechanical stress on ECs, modifying their membranes and the tight junctions between them [64,65,66,67]. The transient disruption induced by microbubble-assisted FUS lasts for a short period of time, from 4 to 6 h, the length of which can be modulated by tuning pulse duration, pulse frequency, exposure time and microbubbles’ diameter [64,65,66,67]. The success of trans-BBB delivery following a microbubble-assisted FUS BBB-disruption event was demonstrated for a broad spectrum of therapeutic materials, including liposome-encapsulated and antibody-based anticancer agents [64,68,69]. Orthogonal techniques, such as magnetic resonance imaging (MRI), have been used to visualize and monitor the process in real-time to ensure adequate optimization of the BBB disruption parameters and deep tissue targeting with efficient spatial resolution [64,65,66,67]. Minimizing aberrations in the phase and amplitude of FUS waves caused by the skull acoustic impedance remains the major challenge of this approach [64,70]. Correction for FUS beam aberrations introduced by the skull, whose thickness varies amongst species, requires determination of the acoustic properties of the test subjects’ skull. This can be achieved by CT scanning, and integrating the resulting data in aberration correction algorithms [64,70]. Correction of abrasions, together with the multiple MRI acquisitions and analysis, makes this approach time consuming, and considering the amounts of both ultrasound and MRI contrast agents, may not always be cost-effective.

The mechanism by which TJs protein impairment is evoked, including how hyperosmotic or mechanical stress contribute to this impairment, remains unknown. In independent studies, immunoelectron microscopic examination after hyperosmotic and FUS-induced disruption showed both redistribution and loss of immunosignals of TJ proteins in affected brain regions [67,71]. Loss of immunosignals can be attributed to disintegration, denaturation or conformational change in TJs’ protein structure [67,71]. The major concern in transient BBB disruption technologies is that during the therapeutic window, the CNS remains unprotected against blood-borne toxins, such as bacteria and viruses, and blood components that are considered neurotoxic. Therefore, the technique requires careful planning, monitoring and optimization of the therapeutic window to ensure optimal therapeutic efficacy and reduced side effects.

#### 2.1.3. Drug Delivery Using Novel Formulations

Designing drug candidates with balanced physicochemical properties that can simultaneously sustain an adequate pharmacokinetic profile of the drug and conquer the BBB remains a very challenging aspect. Such challenge can be offset, as discussed below, by designing sophisticated formulations that can exploit the innate transcytosis capacity of transporters expressed by ECs of the BBB. This can be achieved by ligation of the therapeutic material to a vector whose binding to an EC transporter can elicit a transcytosis event. EC transporters targeted to fulfill this purpose fall into three major categories: solute carriers, protein and hormone receptors, and negatively charged membrane entities. Although some of these transporters are bidirectional, the flux direction is generally set by the concentration gradient from high intravascular to low intracerebral [72]. Exemplary EC transporters that participate in such delivery modality are briefly reviewed in this section, along with prototypical examples of formulations designed to target them.

##### Solute Carrier-Mediated Transcytosis (CMT)

The detailed mechanism of elicited solute carrier-mediated transcytosis (CMT) is believed to proceed via carrier–vector recognition at the luminal side followed by a conformational change of the carrier from outward to inward-facing states, resulting in the transport of the substrate down its concentration gradient across luminal and abluminal membranes of the ECs (Figure 10) [73]. Solute carrier transporters (SLC) are bidirectional and are expressed at both the luminal and abluminal membranes to ensure the translocation of their substrates into the ECs through the luminal membrane and out through the abluminal membrane, without sequestration within the capillary endothelium [11].

##### Glucose Transporters (GLUTs)

Transport of glucose across the BBB is facilitated by several glucose transporters, of which the most prevalent is the sodium-independent bi-directional transporters family (GLUTs) [74]. Due to their abundance, glucose transporters gained significant attention as a potential drug delivery route across the BBB [75,76]. In pre-clinical studies, glycosyl conjugates, d-glucos-6-yl ester, in particular, of the BBB-impermeable antidepressant 7-chlorokynurenic acid were shown to be more potent as neuroprotective agents against seizures induced by N-methyl-d-aspartate (NMDA) in mouse models (Figure 11) [77,78].

Similarly, glycosylation of l-DOPA, an FDA approved dopamine metabolic precursor used for the management of Parkinson’s disease symptoms, was shown to significantly reverse the reserpine-induced hypolocomotion (Figure 11) [79,80]. In contrast, attempts to deliver glycosylated chemotherapeutic reagents across the BBB failed by comparison. Not only were glycosyl conjugates of structurally larger molecules—such as chlorambucil, a chemotherapy medication used to treat chronic lymphocytic leukemia—not transported by GLUTs, but they also inhibited the uptake of ^14^C-labelled glucose (Figure 11) [81]. Albeit reversible, the inhibitory activity of glycosyl conjugates of chlorambucil deemed this approach limited by its high selectivity toward a specific molecular size [81].

##### Large Neutral Amino Acid Transporters (LATs)

The large neutral amino acid transporters (LATs) are transporters associated with the translocation of neutral amino acids that contain relatively large and lipophilic substituents on the α-carbon, such as l-tyrosine and triiodothyronine, across biological membranes [82,83,84]. Pharmacologically significant LATs substrates include the prodrug l-DOPA, used as dopamine replacement in Parkinson’s disease therapy, melphalan, used as a chemotherapeutic agent, and gabapentin, prescribed as anticonvulsant medication (Figure 12) [83,84]. By analogy with GLUTs, LATs are highly selective toward the substrate’s size, conformational rigidity and regiochemistry of its substituents, which can limit this drug delivery approach to a small group of therapeutic materials [83]. Additionally, LATs were found to competitively bind to some of its substrates, which can inhibit their ability to transport other essential amino acid nutrients [83,84].

##### Organic Anion Transporting Polypeptide (OATPs)

Organic anion transporting polypeptide (OTAPs) are protein transporters that mediate the transport of a wide variety of endogenous amphipathic organic substrates, including steroid conjugates, thyroid hormones, bile salts, and oligopeptides [85]. A growing body of pre-clinical studies demonstrated OTAPs’ capacity to transport oligopeptides, such as DPDPE and deltorphin, which bind to opioid receptors expressed throughout brain, spinal cord and peripheral nerves, inducing an analgesic effect against pain secondary to inflammatory pathological conditions (Figure 13) [86,87].

##### Receptor-Mediated Transcytosis (RMT)

In receptor-mediated transcytosis (RMT), a receptor–vector recognition event occurs at the luminal side (Figure 14) [88]. This first point of entry is followed by endosome formation, which pinches the luminal membrane and transports the cargo across the endothelial cell to the opposite end [88]. Transcytosis ends with exocytosis of the cargo at the abluminal side and recycling of the receptor back to the luminal surface of the membrane, free to engage in another RMT cycle [88]. The limiting step in achieving an effective delivery of neurotherapeutics across ECs is the prevention of lysosomal sorting, leading to degradation of the neurotherapeutic cargo.

##### Insulin Receptor (IR)

The insulin receptor (IR) facilitates the transport of insulin, produced nearly exclusively by the pancreas, across the BBB so it can contribute in an array of vital signaling pathways [89]. Engineered monoclonal antibodies with a binding affinity towards the IR can be used to ferry therapeutic materials across the BBB via IR-mediated transcytosis. Erythropoietin (EPO), a neurotrophic factor that supports the growth, survival, and differentiation of neurons, is a protein whose development as a potential therapeutic material for treatment of neurological disorders is restricted by its inability to traverse the BBB [90,91]. Re-engineering of EPO by its genetic fusion to a peptidomimetic monoclonal antibody with an affinity to human insulin receptor (HIRMAb) was shown to markedly increase its brain uptake after intravenous injection into adult rhesus monkeys relative to the native EPO control (Figure 15) [91,92].

##### Transferrin Receptor (TfR)

The transferrin receptor (TfR) is a transmembrane glycoprotein, whose function is to mediate the cellular uptake of Fe-bound Tf [93]. Tf is a member of Fe-binding glycoproteins family and a protein of vital importance for the proliferation, growth, and differentiation of cells [93]. In preclinical studies, anti-TfR monoclonal antibody drug conjugates, such as anti-TfR antibody-methotrexate, a chemotherapeutic agent, were shown to exhibit an enhanced uptake into the brain parenchyma relative to their antibody-free analogues (Figure 16) [94]. The versatility of this approach encompasses not only the delivery of small molecules but also offers a potential delivery route for peptides and neurotrophic factors. Owing to its enhanced transport across the BBB, an anti-TfR monoclonal antibody-EPO conjugate intravenously injected into mouse stroke models dramatically reduced the hemispheric stroke volume compared to its antibody-free counterpart [95].

##### Low-Density Lipoprotein Receptor (LDLR) and Low-Density Lipoprotein Receptor-Related Protein (LPR)

The low-density lipoprotein receptors (LDLRs) and LDLR-related proteins (LRPs) are multifunctional members of the LDLR superfamily, which is engaged in a wide range of physiological processes, including the regulation of cholesterol homeostasis by receptor-mediated uptake of cholesterol-transporting low-density lipoproteins [96]. Ligands that can stimulate LDLs and LRPs-mediated transcytosis across the BBB, such as angiopep, were derived from LDLRs and LRPs substrates, such as amyloid precursor protein (APP) and aprotinin, in particular their Kunitz protease inhibitor domain [97,98]. An angiopep-paclitaxel conjugate (ANG1005) was shown to be significantly more effective at bypassing the BBB and accumulating in brain parenchyma, compared to the free paclitaxel, after its intravenous injection into brain-metastasized breast cancer mouse models (Figure 17) [99].

##### Neurotropic Virulence Factor Receptors (VFRs)

Virulence factors are molecules produced by a pathogen to allow it to enter, replicate, and persist in a host organism’s body [100]. A prominent example is the neurotropic rabies virus (RV), which recruits its external glycoprotein (RVG) to bind to the nicotinic acetylcholine receptor, membrane fusion and host-cell entry [101,102,103,104,105]. The well-characterized mechanism of RV cellular entry inspired the development of a CNS-drug delivery motif, which utilizes RVG-derived peptides as the delivery vector [101,102,103,104,105]. In preclinical gene silencing studies, GFP transgenic mouse models were intravenously injected with GFP-specific small interfering RNA (siRNA) non-covalently conjugated to a chimeric RVG fragment peptide (Figure 18) [103]. Conjugation to RVG fragment peptide enabled the translocation of GFP specific siRNA across the BBB and GFP gene expression was significantly suppressed in the brains of test subjects relative to their other major organs [103].

##### Adsorptive-Mediated Transcytosis (AMT)

Similarly to RMT, adsorptive-mediated transcytosis (AMT) proceeds with an invagination of the AMT-substrate followed by endosomal formation and transport from the luminal to the abluminal side, ending with exocytosis of the endocytosed cargo [47,106,107,108]. Unlike RMT, AMT is triggered by an electrostatic interaction between the AMT-vector and anionic proteoglycans, such as syndecan-4 (Figure 19A) [109,110]. AMT-vectors can be derived from naturally occurring proteins or peptides and are typically short, not exceeding 30 amino acid residues, and feature hydrophobic residues and residues that are positively charged under physiological conditions [111,112,113,114,115,116]. Vectorization by ligation to cell-penetrating peptides, such as SynB_1_, a peptide derived from an antimicrobial peptide isolated from porcine leucocytes, was shown to markedly enhance brain uptake of poor BBB-penetrating drugs, such as cysteamide-modified dalargin, doxorubicin, and benzylpenicillin (Figure 19B) [113,117,118,119]. The most attractive attribute of drug delivery facilitated by AMT is that it does not interfere with endogenous cellular functions the way CMT or RMT-mediated drug delivery do. To date, there are no records, to our knowledge, that can demonstrate an interference of AMT with cellular signaling pathways. Additionally, binding sites are saturated at higher concentrations in AMT than they are in CMT and RMT [38,91,120,121].

##### Nanoscopic Formulations for the Delivery of Therapeutics to the Brain

Nano-formulations have emerged as platforms that allow the modification of the pharmacokinetic or pharmacodynamic properties of therapeutic materials without dramatically modifying their chemical structure (Figure 20). Because clearance of therapeutic material by excretion is dependent on its molecular weight, nano-formulations are designed to increase the apparent therapeutic material’s hydrodynamic volume, reducing its rate of excretion [4,122]. Nano-formulations are also designed to mask the therapeutic material’s functional groups detectable by the host’s immune system, serum proteins and enzymes, prolonging its circulation half-life [122,123,124,125].

Some nano-formulations are engineered with an intrinsic capacity to deliver therapeutics across brain barriers, such as SGT-53, while others are engineered to deliver therapeutics with the aid of recently developed techniques that transiently disrupt the BBB, such as Doxil^®^ (Figure 21) [69,126]. Nano-formulations of several therapeutics have been approved by the Food and Drug Administration (FDA) and are commercially available, such as Doxil^®^ and Onivyde^®^, and some are currently under preclinical and clinical trials, such as NU-0129 and Onzeald^TM^ [127,128].

Developing drug delivery strategies that can overcome the BBB and selectively target the mitochondria is of great interest, since mitochondrial dysfunction has been implicated in the pathology of several neurodegenerative diseases, such as Alzheimer’s disease, Parkinson’s disease, and cancer [129,130,131,132]. Mitochondrial targeting ligands exploit the mitochondrial inner negative potential to stimulate mitochondrial uptake [133,134,135]. Mitochondrial-targeting ligands share similar structural features with AMT ligands. They are small, hydrophobic and positively charged molecules [133,134,135]. One of the important questions we look forward to answering is if the positive charge of mitochondrial-targeting ligands delocalized over their hydrophobic phenyl rings can elicit an AMT-like transport of mitochondrial-targeting conjugates or whether mitochondrial-targeting conjugates require the use of orthogonal techniques that can transiently disrupt the BBB, such as MB-FUS. We set out to test these hypotheses by designing nanoscopic polymeric brush-like architectures (NPBAs) conjugated with mitochondrial-targeting ligands (Scheme 1). Our group prepared a fluorescent mitochondrial-targeting NPBA prototype via Ring Opening Metathesis Polymerization technique (ROMP).

ROMP initiated by Grubbs third generation ruthenium-benzylidene catalyst has been regarded as a powerful tool for the synthesis of graft-through nanoscopic brush architectures with a predictable number average molecular weight, Mn, and a narrow molecular weight distribution, Ð (Scheme 1; Box 1) [136,137]. Triphenylphosphonium ligands have emerged as economically and synthetically accessible mitochondrial-targeting systems that can be accessed from alkyl-halides via a substitution reaction using PPh_3_ as the nucleophile in a pre-polymerization step [138,139].

A range of commercially available hydrophilic polymers, such as chitosan, polyethylene glycol and poly(N-(2-hydroxypropyl)methacrylamide), can be used for the hydrophilic skeleton design, but because polyethylene glycol has only two terminal functional groups, its functionalization is practically more precise to accomplish and monitor [122]. Additionally, the ethylene glycol oligomers can be prepared in-house in good yields and with high oligomeric purity via oligomerization of tetraethylene glycol [122,140]. The diethylamino coumarin derivative proved useful as a fluorescent tag due to its two-photon fluorescence, which can be induced via illumination with a stream of a pulsed near-infrared laser beam with a pulse duration and repetition rate of 100 fs and 80 MHz, respectively [122,141,142,143]. The optical consequence of two-photon fluorescence is the confinement of the fluorescent region to a smaller volume than that of conventional one-photon fluorescence [122,141,142,143]. In practical terms, this leads to suppression of out-of-focus fluorescence, which enhances the resolution of the image and minimizes photobleaching of fluorophore molecules and photochemically-induced cytotoxicity of the live specimen [122,141,142,143,144,145]. Nonetheless, the optical probe is a bright UV-Vis, excitable fluorescent dye that can generate good images with a conventional confocal microscope [122,141]. Preliminary mitochondrial-uptake assessment of NPBA-I conducted employing a monolayer of SH-SY5Y cells showed promise but more effort is still needed to bring this idea to its full potential.

Box 1Advantages of ROMP in the Context of drug-delivery formulation design.The beauty of ROMP is its ability to integrate the mitochondrial targeting ligand or the fluorescent tag, or any other desirable functional groups, orthogonally to the polymerization process. If ROMP was conducted under living polymerization conditions, the desirable functional groups are controllably and homogenously incorporated into polymer molecules in a bottom-up approach. Use of nanoscopic formulations with homogenous distribution of the desired functional group(s) in biological experiments generates reproducible data, creating more opportunity for new insights.

### 2.2. Challenges in Translation from Pre-clinical Evaluation to Clinical Use

After toxicity and efficacy of drug delivery formulations are assessed in animal models, human trials are initiated, the outcome of which can be communicated to request an FDA approval for marketing purposes. Clinical trials are broken into three main stages that build on one another: a dosing, toxicity and excretion in healthy subjects stage (phase I); a safety and efficacy in a larger group of patients with the target illness stage (phase II); and a multi-center, randomized and placebo-controlled stage (phase III) [146]. We surveyed the clinical trial database available at the ClinicalTrials.gov repository in April 2019, bearing in mind that the status of trials changes over time, using search terms centered around the BBB. We cross-checked the outcome with recently published reviews in the area of drug delivery formulations to assess the progress in the field of drug delivery to the CNS [147,148]. We listed a glimpse of the survey findings, which we believe is of relevance to the scope of this review (Table 1). A number of drug delivery formulations that were designed to deliver therapeutics to the CNS by exploiting the innate transcytosis capacity of ECs have been granted approval to undertake clinical trials. The transition of CNS-targeting formulations, such as ANG1005 and SGT-53, from preclinical investigations to the clinical trials pipeline is indicative of the feasibility of transporter-mediated transcytosis approach. Nonetheless, the scarce number of successful clinical trial transitions of transporter-mediated transcytosis formulations suggests that the extrapolation of data generated from animal studies to humans is encountering significant challenges. Thus, what are the key challenges in CNS-targeted drug delivery development that can delay the translation of preclinical findings to clinical use?

#### 2.2.1. Safety Liabilities due to Nonspecific Body Distribution

In freshly isolated human erythrocytic cells, glycosyl conjugates of chlorambucil **8** and **9** were shown to inhibit the uptake of d-[^14^C]glucose in a concentration-dependent manner [81]. The interaction of glycosylated chlorambucil esters **8** and **9** with the GLUTs was concluded to be in the form of a non-transported inhibitor [81]. In an independent study, mice treated with TfR-MAb suffered from a marked decrease in immature reticulocyte count relative to mice treated with non-immune IgG control [179]. Reticulocytes develop into red blood cells through erythropoiesis; a process that requires maintaining iron homeostasis [180]. The indiscriminate uptake of transporter-mediated transcytosis formulations was reported in several pre-clinical studies to extend beyond constituents of the circulatory system to encompass major body organs such as spleen and liver which is conceivable due to ubiquitous expression of protein transporters in several body regions [91,118,119,181,182,183,184,185,186,187,188,189,190,191,192,193,194,195]. These phenomena raise efficacy and safety concerns associated with the peripheral uptake of these formulations and their competition with natural substrates for the binding to carrier or receptor proteins. It is clear thus far that transporter-mediated transcytosis approach can improve brain uptake of the therapeutic payload. However, consequent to their indiscriminate uptake profile among major body organs, can transporter-mediated transcytosis formulations, under the currently employed experimental conditions, allow the therapeutic payload to consistently and robustly attain a therapeutically effective concentration in the brain? If so, would the aforementioned transient nutritional deprivation be a tolerable compromise? Alternatively, how can the chemical structure of the transporter-mediated formulation be tailored to find a good balance between efficacy and safety? 

#### 2.2.2. Inadequate Endothelial-Parenchymal Transport

Brain permeability under saturable uptake processes such as transporter-mediated transcytosis is often described by Equation (2) where PS is the brain permeability surface, C_Pf_ is the concentration of test solute in the perfusate, V_max_ is the maximal rate of the saturable component of transport, K_m_ is the perfusate concentration at which half maximal transport occurs, and K_d_ is the non-saturable diffusion uptake clearance [39]. Based on this conception, drug-conjugates had been formulated to possess high affinity, therefore low K_m_ values, towards the target protein transporter [91,121]. However, recent findings emerged to contradict with this conception revealing that high affinity towards the protein transporter, such as TfR, that is not easily reversed may obstruct the release of the drug-conjugate from the abluminal membrane which is the final stage of the transcytosis pathway [196]. In parallel preclinical studies employing in vitro models, strong binding to TfR was shown to trigger lysosomal sorting which may contribute to the obstructed transcellular trafficking observed in vivo [197,198,199]. These findings invalidate previous communications and invite researchers in this field to; (1) Study structure-function relationship of drug-conjugates in more detail in the context of transport across the BBB to brain parenchyma, and (2) Critically analyze their brain uptake assessment protocols in an attempt to account for such discrepancies.
(2)$$PS=\frac{V_{max}}{K_{m}+C_{pf}}+K_{d}$$

#### 2.2.3. How We Measure Drug Transport across the BBB

Brain uptake of a test substrate via a specific transport mechanism can be estimated in animal models through behavioral assays or radiolabeling experiments. **Behavioral assays** are used to measure brain permeability as a function of the potency of an intravenously or intraperitoneally injected test substrate at reducing the severity of, delaying the onset to or reverting symptoms induced by exposure to noxious stimuli [77,78,79,95,117]. **Radiolabeling experiments**, which is the more conventional method, directly measure brain uptake of a radiolabeled test substrate relative to a differentially-radiolabeled reference using a liquid scintillation counter [83,87,91,94,99,117]. The differentially-radiolabeled reference can be BBB-impermeable, the use of which corrects for artifactual BBB permeability induced under experimental conditions or during brain tissue sampling and processing [94,117,196,197]. To assess the nature of the transport mechanism of the test substrate, whether it’s competitive or non-competitive, a differentially-radiolabeled and BBB-permeable agent transported via the same mechanism as the test substrate is used as a second reference [83,84]. Radiolabeling experiments employ complementary techniques, such as capillary depletion, to distinguish the fraction of the test substrate that has successfully bypassed the BBB from the fraction that has been sequestered by ECs. The concentration of the test substrate accumulated in brain tissue, C_br_, can be calculated using Equation (3), where C_p_(T) designates blood or plasma concentration at the sampling time T, by fitting the experimentally determined distribution volume of the test substance, V_D_, and reference, V_0_ [39].
(3)$$C_{br}=\left(V_{D}-V_{0}\right)\times C_{p}\left(T\right)$$

However, data generated from radiolabeling experiments should be interpreted with caution as multiple factors involved in the experimental design can lead to erroneous interpretation of results [200,201,202,203,204]. Such factors range from poor recovery of the radioactive material from tissue samples to errors in sample analysis due to quenching of radiolabeled substance during radioactivity measurements [203]. The factor that remains the most detrimental is the degradation of the radiolabeled tracer into radiolabeled metabolites that can traverse brain barriers via alternate transvascular transport mechanisms [203]. This phenomenon leads to overestimation of brain uptake of the parent radiolabeled substance via the scrutinized mechanism which results in discrepancies amongst preclinical findings. Perhaps investigation may be accurately conducted if a metabolic profile of the circulating test substrate and its metabolites is obtained by plasma sample collection at multiple time intervals [203]. Alternatively, the use of chromatographic techniques such as capillary electrophoresis may comprehensively and simultaneously assess brain uptake of the test substrates and its metabolites [202]. The cerebrovascular transport of the scrutinized substrate can be studies in its intact for using in situ brain perfusion technique (Box 2) [83,84,87,99,117].

Box 2Approaches to Reducing Metabolic Liability of Test Substrates.***In situ brain perfusion technique*** was developed to minimize the mixing of the infused test substrate with endogenous plasma, reducing its metabolic degradation [83,84,87,99,117]. In this technique, blood circulation to the brain in a fully anesthetized animal model is taken over by an infusion of oxygenated physiological buffer. This can be accomplished by ligation of external and internal carotid artery branches, direct catheterization of the carotid artery, and right before starting the infusion, the heart ventricles are severed to stop endogenous blood from mixing with the infusate. Brain is perfused for a brief period, typically 60–120 s, with the physiological buffer containing the test and reference substrates, then washed for 30 s to remove unbound material from brain vasculature. Animals are sacrificed, decapitated, and their brains are removed and prepared to assess their content of radioactive substrates. Albeit suitable for studying the kinetics of the unidirectional uptake of intact substrate, this technique does not fully describe what the injected therapeutic material undergoes in a biological environment.

#### 2.2.4. Species Differ in Parameters that control the BBB permeability

Studies employing quantitative-targeted absolute proteomics (QTAP) coupled with Liquid chromatography–tandem mass spectrometry (LC–MS/MS) revealed that the expression level of functional proteins such as efflux pump, SLC and receptor proteins generally varies amongst different species such as rats, mice, monkeys and humans (Box 3) [75,76,205,206]. The differential expression of BBB proteins is paralleled with a varied brain uptake of substrates across different species. Higher brain distribution of radiolabeled P-gp substrates such as ^11^C-verapamil and ^18^F-altanserin was detected in the brains of healthy humans and monkeys compared to rodents [75,207]. Not only the species must be carefully selected before designing an experiment, but also the route of administration. The basis of this selection must be elaborated in the study as the administration route constitutes an important variable that may significantly influence the results (Box 4).

#### 2.2.5. BBB Protein Expression is Altered under Different Pathological Conditions

BBB breakdown has been identified in many patients with various neurological disorders through functional imaging and postmortem analysis. Neuroimaging studies that employ dynamic contrast enhanced-magnetic resonance imaging (DCE-MRI) showed an increase in brain uptake of gadolinium, an MRI tracer, amongst individuals diagnosed with different neurological diseases which is an indicative of capillary leakage (Box 3) [16,208,209,210,211,212,213,214,215,216]. Independent studies using positron emission tomography (PET) reported the detection of a diminished glucose uptake and impaired P-gp function in the brains of AD and PD patients [16,217,218,219,220,221,222,223,224]. Neuroimaging using arterial spin labeling- magnetic resonance imaging (ASL-MRI) revealed that patients diagnosed with neurological diseases suffer from abnormal brain perfusion and reduced cerebral blood flow which can further aggravate BBB damage [225,226,227,228,229,230,231,232,233]. Postmortem analysis of brain samples that belonged to patients diagnosed with different neurological diseases and showed no symptoms of mixed dementia demonstrated perivascular deposits of blood-derived proteins deemed toxic to neuronal function [234,235,236,237,238,239,240,241]. This observation was often accompanied with degeneration of BBB structural components such as ECs, TJs, pericytes and basement membrane along with dissociation of astrocytic end-feet from capillaries [242,243,244,245,246,247,248,249,250,251,252,253,254,255,256]. 

Box 3Outstanding Questions.
How does microvessel composition of carrier proteins, receptors and acidic and neutral lipids in the brain compare to microvessel composition in other major body organs, such as kidneys, liver, spleen and lungs, and how does this vary among different species?What is the significance of pathological BBB breakdown in the context of targeted drug delivery? Can BBB breakdown act as a gateway for therapeutic access?Is the downregulation of transporters activity a phenotypic hallmark of a dysfunctional BBB?How does the alteration of transporters’ activity progress with the aggravation of disease state and how does it vary under different pathological conditions?How would advances in the knowledge surrounding pathological BBB breakdown shape the future of research being conducted to exploit the CMT and RMT capacity of ECs in the context of drug delivery through the BBB?


Box 4Influence of Drug Administration Route.There is a sequence of events that any consumed drug must follow before it elicits its therapeutic effect. Events include administration, release from its physical form, absorption from the site of administration into the body, and finally accumulation at the site of action [257]. The route of administration is the first point of access and the first stage at which the drug bioavailability, a measure of systemic availability of a drug, is dictated. The route of administration and how it contributes to the success of drug delivery to the CNS is not within the scope of this review, crucial though it is, only because we believe that it requires a dedicated review to grant this factor the attention it deserves. We can still emphasize the complexity of coordinating the physicochemical properties necessary for optimal drug pharmacokinetics and pharmacodynamics, which can be further complicated when the route of administration is factored in the drug development process. Drugs administered orally, intravenously, subcutaneously, transdermally, intracarotidly or nasally will encounter different en route factors that can influence their dissolution, their release from their physical form, their stability and their absorption [257]. For instance, in oral administration, which is the most popular route of drug dosing, factors such as digestive enzymes, gastric acid, stomach-emptying rate and intestinal motility regulate the rate of drug absorption, metabolism and disposition via the gastrointestinal tract [257]. Consequently, the ideal analysis of the structure-function relationship must accommodate the various administration route-associated factors, not only the nature of the drug and the formulation design.

## 3. Conclusions

The blood-brain barrier, the meningeal or subarachnoid barrier and the choroid plexus barrier work in consortium to maintain cerebral homeostasis and vital brain functions. Due to its large surface area and its faster blood flow rate, the BBB is considered as the primary contributor to the brain’s strict permeability. The BBB is a continuous layer of endothelial cells fused together by virtue of their tight junction proteins. The BBB is a dynamic assembly whose integrity is modulated in response to neuronal needs. Due to the growing population of patients affected with neurodegenerative diseases, an urgent need for drug delivery strategies that can bypass the BBB has emerged. The progress achieved thus far is divided into surgical approaches that can deliver the therapeutic material beyond the BBB using specialized catheters or implantable pumps, and pharmacological approaches that exploit endogenous transport mechanisms to deliver the therapeutic material through the BBB (Table 2). Surgical approaches are invasive and often fraught with technical complications accompanied with inevitable, and sometimes permanent, damage to nearby tissues, as well as tissue infection. Pharmacological approaches tackle drug delivery through the BBB by either altering the therapeutic material’s physicochemical properties to facilitate its passive diffusion through the LBL, attenuating the efflux power by co-administration of efflux-pump inhibitors, increasing BBB permeability by its transient disruption, or the ligation of the therapeutic material to a vector whose interaction with EC transporters can trigger a transcytosis event.

Altering the physicochemical properties of therapeutic materials is often guided by computational methods derived from the retrospective analysis of CNS and non-CNS active drugs that are commercially available or still undergoing clinical trials. Albeit effective, these methods are still limited by the number of predictive descriptors (lipophilicity, H-bonding, rotatable bonds, etc.), which narrows down the window of drug discovery to encompass a small set of molecules and preclude any serious attempts to explore molecules beyond its limits. The long-term effect of downregulation of efflux-pumps activity, especially in treatment regimens that requires constant and repeated exposure to the inhibitors, is yet to be determined. Similarly, how the inhibition of efflux-pumps ubiquitously expressed in cerebral and peripheral human tissues can alter the pharmacokinetics of the therapeutic material is yet to be unveiled. By the same analogy, the long-term effect of transient BBB disruption, induced by hyperosmosis or microbubble-assisted focused ultrasound, and the resilience of the BBB after repeated and consecutive disruption events remain to be investigated.

Formulations that can elicit a transcytosis event by interacting with EC transporters are designed to target solute carriers, receptors or negatively charged transmembrane proteins. The choice of the target transporter depends on many variables, such as the size and the chirality of the drug-conjugate and whether the ability to mimic the transporter’s natural substrate is essential, or not, to stimulate the binding of the formulation to the target transporter. Sophisticated formulations developed to effectively ferry neurotherapeutics across brain barriers have had promising preclinical success, however, they are still struggling to push their way forth towards clinical use. This brings up an important question—what can we do to facilitate the extrapolation of data generated from animal studies to humans to realize the full potential of this approach?

First of all, information surrounding microvessels composition of carrier proteins, receptors and acidic lipids in major body organs and how it varies under different pathological conditions is sorely lacking. Additionally, preliminary proteomic analysis conducted across multiple species have emphasized species differences in protein expression and suggest that the behavior of test substrates in a human setting may be better predicted in apes than in rodents. The knowledge of microvessel composition and how it varies amongst different species and under different pathological conditions will ultimately guide the optimization of drug-formulation properties to reduce off-site distribution and minimize safety liabilities.

Moreover, development of a range of transporter-specific radioactive tracers that are metabolically stable during the experimental timeframe may further advance the use of neuroimaging technologies, providing more insight into how transporters activity can be altered with age and under different pathological conditions. The library of transporter-specific radioactive tracers needs to be expanded to encompass substrates of various protein transporters—not only GLUTs and P-gp—that can be used to monitor and detect changes in the activity of various transporters.

Furthermore, brain uptake quantification methods need to be thoroughly and critically assessed prior to embarking on a preclinical in vivo experiment. The injected test substrate can undergo numerous elimination pathways, some of which may generate several metabolites. If autoradiography or liquid scintillation counting or any other radio-labeling based techniques is the method of choice, then researchers need to verify the isotopic purity of the radiolabeled tracer prior to its administration, make sure that the radiolabeled isotope does not reside at a metabolically labile position, and scrutinize and establish the metabolic profile of the radiolabeled tracer under the employed experimental conditions using orthogonal techniques, perhaps chromatographic, to account for the presence of any radiolabeled metabolites. Standardizing experimental parameters is imperative to avoid any discrepancies resulting from overestimation of brain uptake of the radiolabeled tracer.

Transvascular transport to brain lesions across the BBB is a unique process compared to drug transport to lesions in peripheral tissues. It requires the drug-conjugate to successfully bypass the luminal membrane, traverse the intracellular space evading lysosomal sorting and subsequent degradation, and finally bypass the abluminal membrane, whilst maintaining the viability of the transporting cells. In depth analysis of the structure-function relationship in the context of transvascular transport and what it entails of transporter protein binding and dissociation kinetics is a prerequisite for a successful drug-conjugate design, and pre-clinical studies that investigate this relationship are scarce.

The transcytosis-mediated transport approach may be regarded by some as an unfulfilled fantasy, yet the idea, and the attempts generously dedicated to realizing it, have advanced our knowledge of transvascular drug delivery and drug-formulation engineering and how the structure of delivery vectors can be tailored to evade lysosomal sorting, achieving a complete and successful transcytosis event. Therefore, because of the success achieved thus far and the potential room for improvement in the way we plan and execute our in vivo experiments, it is premature to deem this approach infeasible.

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
