# Peer review of "Approaches to CNS Drug Delivery with a Focus on Transporter-Mediated Transcytosis"

_ijms, 2019, doi:10.3390/ijms20123108_

Reviewer 1 Report

The paper entitled "Approaches to CNS Drug Delivery with a Focus on Transporter-Mediated Transcytosis" by Razzak and collaborator is a very nice review describing all most recent advances in central nervous system drug delivery. I really enjoyed reading the paper. It is well written and organised. My recommendation is minor revision. In particular, 

1) Given this is an extremely comprehensive review, I suggest the authors to include a table which summarises delivery strategies with pros and cons.

2) I would clarify that proteins in Fig. 6 are not in scale. The graphical representation of the adsorption case is not clear.

3) In Fig. 7, I would report what the pink symbol represents (i.e. the drug).

4) I would discuss whether or not there are administration differences between small molecules drugs and protein therapeutics. 

5) I would also add a box about the relationship between physical-chemical properties of drugs and administration strategies.

Author Response

1) Given this is an extremely comprehensive review, I suggest the authors to include a table which summarises delivery strategies with pros and cons.

Response: Suggestion was noted and review was amended accordingly

2) I would clarify that proteins in Fig. 6 are not in scale. The graphical representation of the adsorption case is not clear.

Response: Suggestion was noted and review was amended accordingly

3) In Fig. 7, I would report what the pink symbol represents (i.e. the drug).

Response: Suggestion was noted and review was amended accordingly

4) I would discuss whether or not there are administration differences between small molecules drugs and protein therapeutics.

5) I would also add a box about the relationship between physical-chemical properties of drugs and administration strategies.

Response: (For both 4 and 5, Suggestion was noted and review was briefly amended because drug administration route is an important concept that may need a dedicated review to explain its ramification on CNS drug delivery)

Reviewer 2 Report

Fantastic work!!

Only several minor recommendations:

(1) Section 1.2.2.4: There are many glucose transporters or amino acid acid transporters involved in BBB drug delivery. It's ideal to use plural i.e. GLUTs, LATs or OAPTs. Or use specific transporter name such as LAT1 (SLC7A5), etc.

(2) There are too many layers of category. It will be very confusing to the readers. I would not use bold font for all sub-categories. Maybe also consider using other numbering system, such as :

1.2.2-a, 1.2.2-1a, or 1.2.2 (I), II, III etc.

(3) Some minor typo need to be corrected i.e. (AUC0T on page 6, glycine-NMDA on page 12).  

 Author Response

(1) Section 1.2.2.4: There are many glucose transporters or amino acid acid transporters involved in BBB drug delivery. It's ideal to use plural i.e. GLUTs, LATs or OAPTs. Or use specific transporter name such as LAT1 (SLC7A5), etc.

 Response: Suggestion was noted and review was amended accordingly

 (2) There are too many layers of category. It will be very confusing to the readers. I would not use bold font for all sub-categories. Maybe also consider using other numbering system, such as :1.2.2-a, 1.2.2-1a, or 1.2.2 (I), II, III etc.

Response: Suggestion was noted and review was amended accordingly

 (3) Some minor typo need to be corrected i.e. (AUC0T on page 6, glycine-NMDA on page 12).

Response: Typographical errors were noted and corrected. Review was screened for additional errors.  

Int. J. Mol. Sci. EISSN 1422-0067 Published by MDPI AG, Basel, Switzerland RSS E-Mail Table of Contents Alert
Back to Top